# GELLO: A General, Low-Cost, and Intuitive Teleoperation Framework for Robot Manipulators

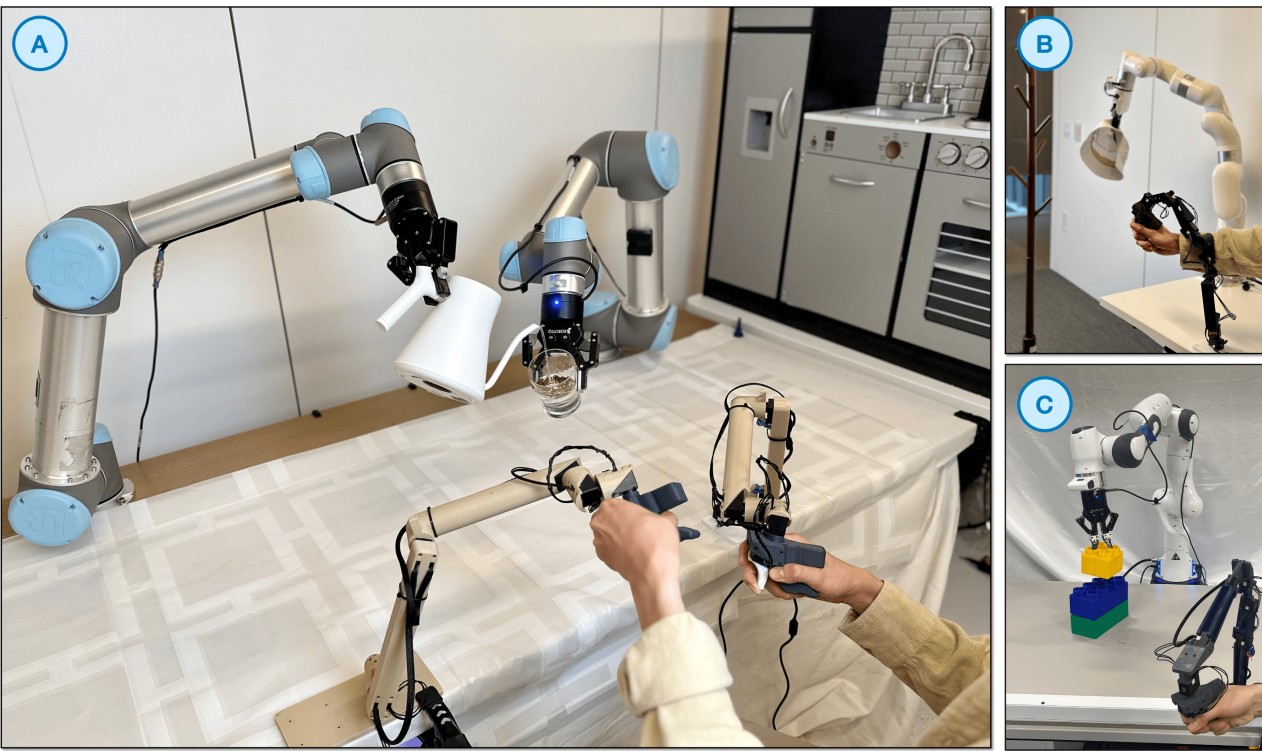

**Fig. 1:** We show GELLO teleoperation systems built for three different types of robots: (A) two UR5s, (B) an xArm, and (C) a Franka. The user can teleoperate the robot arms by controlling the GELLO devices. The bill of materials for each GELLO device is less than $300.

*Abstract*— **Imitation learning from human demonstrations is a powerful framework to teach robots new skills. However, the performance of the learned policies is bottlenecked by the quality, scale, and variety of the demonstration data. In this paper, we aim to lower the barrier to collecting large and high-quality human demonstration data by proposing GELLO, a general framework for building low-cost and intuitive teleoperation systems for robotic manipulation. Given a target robot arm, we build a GELLO controller that has the same kinematic structure as the target arm, leveraging 3D-printed parts and off-the-shelf motors. GELLO is easy to build and intuitive to use. Through an extensive user study, we show that GELLO enables more reliable and efficient demonstration collection compared to commonly used teleoperation devices in the imitation learning literature such as VR controllers and 3D spacemouses. We further demonstrate the capabilities of GELLO for performing complex bi-manual and contact-rich manipulation tasks. To make GELLO accessible to everyone, we have designed and built GELLO systems for 3 commonly used robotic arms: Franka, UR5, and xArm. All software and hardware will be open-sourced.**

## I. INTRODUCTION

In recent years, robotics has been through a remarkable transformation driven by the increasing integration of data-driven methods in every component, ranging from perception to control. The ability to learn from diverse data helps robots to generalize in wide scenarios which would be difficult to achieve by a manually designed system. For robotic manipulation, there has been great successes by learning from human demonstration data [1], [2]. More recently, we have seen that system performance continues to increase with larger scale demonstration datasets and can even generalize to open vocabulary language instructions [3], [4], [5], [6]. However, existing systems are bottlenecked by the dataset size and the complexity and diversity of the tasks humans can perform through a teleoperation system.

For manipulation tasks, commonly used teleoperation systems capture control signals from input devices like 3D mouse [7], VR controllers [8], [9], or cameras, these systems abstract away the kinematic constraints on the robot and can be unintuitive to new users. On the other hand, robotics has a long history of teleoperation through bi-lateral devices [10], where operators receive feedback, but are generally more costly. More recently, the ALOHA system, although unilateral, presents impressive teleoperation capabilities for fine-grained manipulation tasks with low-cost hardware built using off-the-shelf robot arms [11]. Nevertheless, the ALOHA system is tailored to a specific robot arm and has a higher cost due

to having additional robot arms as controllers for the user.

In this paper, we introduce GELLO, a **GE**nera**L**, **LO**w-cost, and intuitive teleoperation framework for robot manipulators. GELLO is designed to be low-cost, easy to build, and intuitive for humans to use. The key idea is to build miniature, kinematically equivalent controllers with 3D-printed parts and off-the-shelf motors as joint encoders. We clarify that the ideas behind GELLO are not new, rather our contributions can be summarized in the three points below:

1) We present practical implementations of GELLO as a teleoperation system for three commonly used robot arms with simple and low-cost designs.
2) We perform a comprehensive user study demonstrating the system's effectiveness compared to other prevalent low-cost teleoperation systems in the literature.
3) We open-source the hardware and software needed to replicate and operate GELLO to ensure accessibility[1].

## II. RELATED WORK

### A. Teleoperation Systems for Manipulation

**Low-cost Controllers.** Teleoperation systems have a long-standing history and various low-cost sensors have been used to provide the interface for human-robot interaction. Commonly used teleoperation systems include joysticks and spacemouses [12], [13], [14], commercial VR controllers [15], [16], [17], [18], RGB cameras [19], [20], [21], [22] or IMU sensors [23], [24], [25], However due to the morphological differences between these control devices and the robots, the user often can only perform teleoperation in the more abstracted end-effector space. The kinematic constraints of the robot arms therefore are not perceived by the human operators. This prevents the operator from precisely controlling the arm near the areas of kinematic singularities, and self-collisions, which reduces the demonstration throughput and increases failures. Moreover, both VR and camera-based solutions can suffer from occlusion and additional latency.

Notably, the recent ALOHA system showcases impressive fine-grained bi-manual manipulation with Dynamixel-based servo arms, where an additional two arms function as controllers [11]. Similar to other more conventional but costly teleoperation systems such as that of [26], [27], [28], the teleoperation device is another fully-fledged robot arm, with size and capability comparable to the manipulator arm, which can increase the cost. In comparison, with GELLO, we use low-cost components to design a scaled replica of the target arm, resulting in an economical solution that still maintains the advantages of using a kinematically isomorphic arm as the controller.

**Bilateral Teleoperation Systems.** In contrast to unilateral teleoperation approaches, bilateral teleoperation enables the user to feel force feedback from the target arm. This is an active area of research, with a wide range of methodologies. One such approach uses additional robot arms that are isomorphic to the target robots serving as controllers [26], [27], [28], [29], [30], [31]. This approach enables environment

[1]Website `https://wuphilipp.github.io/gello/`

**TABLE I:** A cost comparison of the price of commonly used teleoperation systems. We will show that GELLO compares favorably to other low-cost options (spacemouse and VR) in our user study while being orders of magnitudes cheaper than other options.

| Teleop Device | Approximate Cost |
|---|---|
| 3D Mouse (SpaceNavigator[7]) | $150 |
| GELLO (Ours) | $300 |
| VR (Meta Quest 2) [9] | $300 |
| Robot-to-robot Teleop (e.g. UR5) | $30,000 |
| Haptic Device (Omega7 [36]) | $40,000 |

feedback from the manipulator to be directly relayed back to the joints of the control device. Also, the user can easily sense the kinematic constraints of the robot arm as the controller arm has the same kinematic constraints. Additionally, there have been efforts to design 1-to-1 exoskeletons for teleoperation purposes [32], [33], [34]. These systems, though effective, are typically bespoke for specific robots, leading to a varied design approach across different robot types. Another avenue explored in the literature is the use of special input devices with haptic feedback [35], [36]. While these devices offer a tangible sense of the robot's kinematic constraints, they often have a very tight operation space and additionally require translating the robot's kinematic constraints into tangible force feedback increasing system complexity. In contrast to these approaches, our contribution presents a generalized framework for designing affordable and easily accessible exoskeleton-like unilateral controllers. We provide instances of our approach for three widely-used robot arms. Our system, GELLO, stands out for its affordability, portability, and replicability, reducing the challenges associated with collecting quality teleoperated human demonstrations.

### B. Learning from Human Demonstrations

Learning from demonstrations has been a popular framework for enabling robots to perform a wide range of tasks [3], [4], [5], [15], [22], [37], [38]. Prior works have observed that the performance of the learning system scales with the size of the dataset. As such, there are substantial ongoing efforts at collecting larger and larger datasets [39], [40]. However, collecting human demonstrations can be expensive and time-consuming. For example, the data collection process as done in [3] spanned over 17 months with a team of researchers. On the other hand, significant efforts have been made towards better human-robot interaction to address the bottleneck. Some examples include sharing control between the human and the robot [12], or enabling a human to operate multiple robots simultaneously [41]. These approaches are complementary to our objective, which is to build teleoperation systems that are more accessible and intuitive to use. Finally, a promising direction is learning directly from human video data [42], [43], [44], [45], [46]. While collecting videos of humans performing the tasks directly is relatively inexpensive, overcoming the morphology gap between robots and humans remains challenging for policy learning.

## III. TELEOPERATION DEVICE DESIGN

The focus for the design of GELLO is to create an interface that is both economically accessible and easy-to-use for users aiming to render high-quality demonstrations in robot learning. The primary design principles are summarized as follows:

**Low-cost:** We aim to show that a capable system can be constructed at an affordable price, thus minimizing the entry barrier. This is achieved through the use of economical backdrivable servo motors, 3D printed components, and a minimalist design, making it possible to construct a teleoperation solution for under $300. A cost comparison is shown in Table I. We will show that GELLO outperforms other low-cost options in our user study while being much cheaper than the other systems.

**Capable:** GELLO is designed to be easy to use for human operators. We demonstrate GELLO's capabilities on a range of complex bi-manual manipulation and contact-rich tasks.

**Portable:** Diversity of demonstration data collected in different tasks and environments is critical to the final performance of the learning system. As such, we design GELLO to be compact, and self-contained, facilitating easy transportation. We show GELLO performing tasks across both lab and in-the-wild environments.

**Simple to replicate:** The sourcing for parts is minimal beyond off-the-shelf motors and 3D printing components. The assembly process is also straightforward, requiring minimal technical expertise.

We follow these design principles to make GELLO for three types of robot arms, as shown in Figure 1. The instantiation of our approach centers around critical components like motor selection, kinematically equivalent structure, 3D printed parts, and gravity compensation.

*a) Servo selection:* The critical component to enable GELLO's construction is the availability of low-cost, fully-featured servos. Specifically, we used the DYNAMIXEL XL330 series [47]. Despite their affordability, these servos are equipped with high-resolution 12-bit encoders. These encoders provide measurements of the servo's position, allowing accurate mapping of the controller's configuration to the target arm. In principle, a servo is not even necessary for the construction of GELLO, as we only need to read joint positions. However, in practice, a servo package provides an easy off-the-shelf, self-contained solution that has an encoder and communication protocol, simplifying construction, usage, and maintenance. In addition, the servo actuator provides resistance as the user backdrives it, which acts as natural damping and improves stability for the user.

*b) A scaled kinematically equivalent structure:* We build GELLO as a small-scale version of the target arm which possesses a kinematically equivalent structure. This means that the joints and links of GELLO correspond directly to those of the target arm, allowing the user to control the GELLO manipulator as if they were directly controlling the target arm, as in kinesthetic teaching [48]. Target joint positions are directly sent to the target arm for operation, avoiding the need to compute inverse kinematics. The user can feel resistance from the controller when the joints are close to kinematic

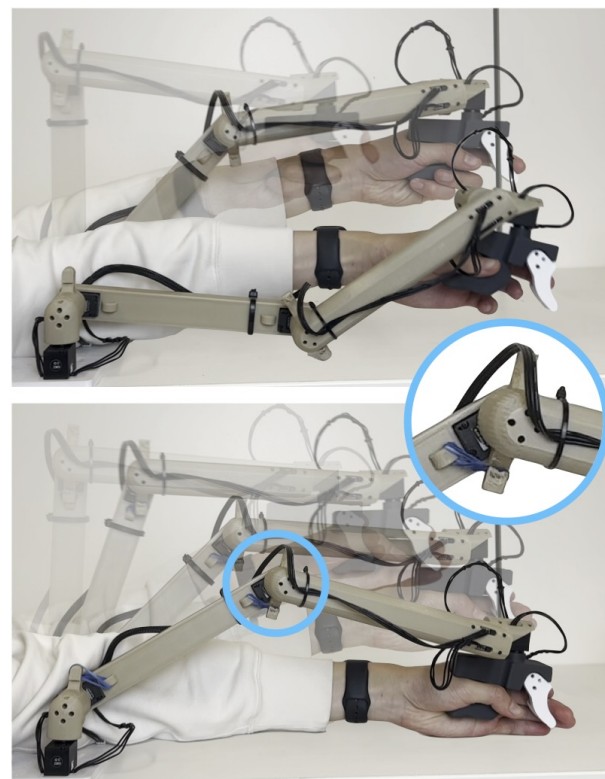

**Fig. 2:** This figure illustrates the trajectory of GELLO both with and without gravity compensation. Top: Without gravity compensation the elbow joint drops. This results in an unfavorable joint configuration for subsequent tasks or even collision with the table. Bottom: With gravity compensation in place, the elbow exhibits minimal movement, leading to a more advantageous joint configuration. We find that simple use of rubber bands or springs are effective.

singularities or joint limits and is thus more aware of these failures, leading to more reliable teleoperation. At the same time, the miniature design makes the controller more portable while still allowing the user to operate full-scale robot arms.

*c) 3D printed parts:* The use of 3D printed parts in GELLO allows a high degree of customization, enabling users to design and print parts that match the specific robot hardware. 3D printing allows us to easily design GELLO systems for 3 kinematically different robots. 3D printing is also a cost-effective method of producing parts, further contributing to the low-cost nature of GELLO.

*d) Gravity compensation:* Gravity compensation is a necessary component of GELLO's design. It counteracts the force of gravity on the manipulator, making it easier for the user to control. We employ rudimentary but effective passive gravity compensation that involves the use of mechanical components such as springs or rubber bands to offset the weight of the manipulator. These components also acts as a joint control regularizer. It ensures that the arm maintains a "natural" posture. This prevents the arm from adopting other kinematically viable yet unconventional positions, as illustrated in Figure 2. We only add gravity compensation to the joints that exhibit the most significant resistance against gravity in the arm's default resting position, which for the UR design, is the second and third joint.

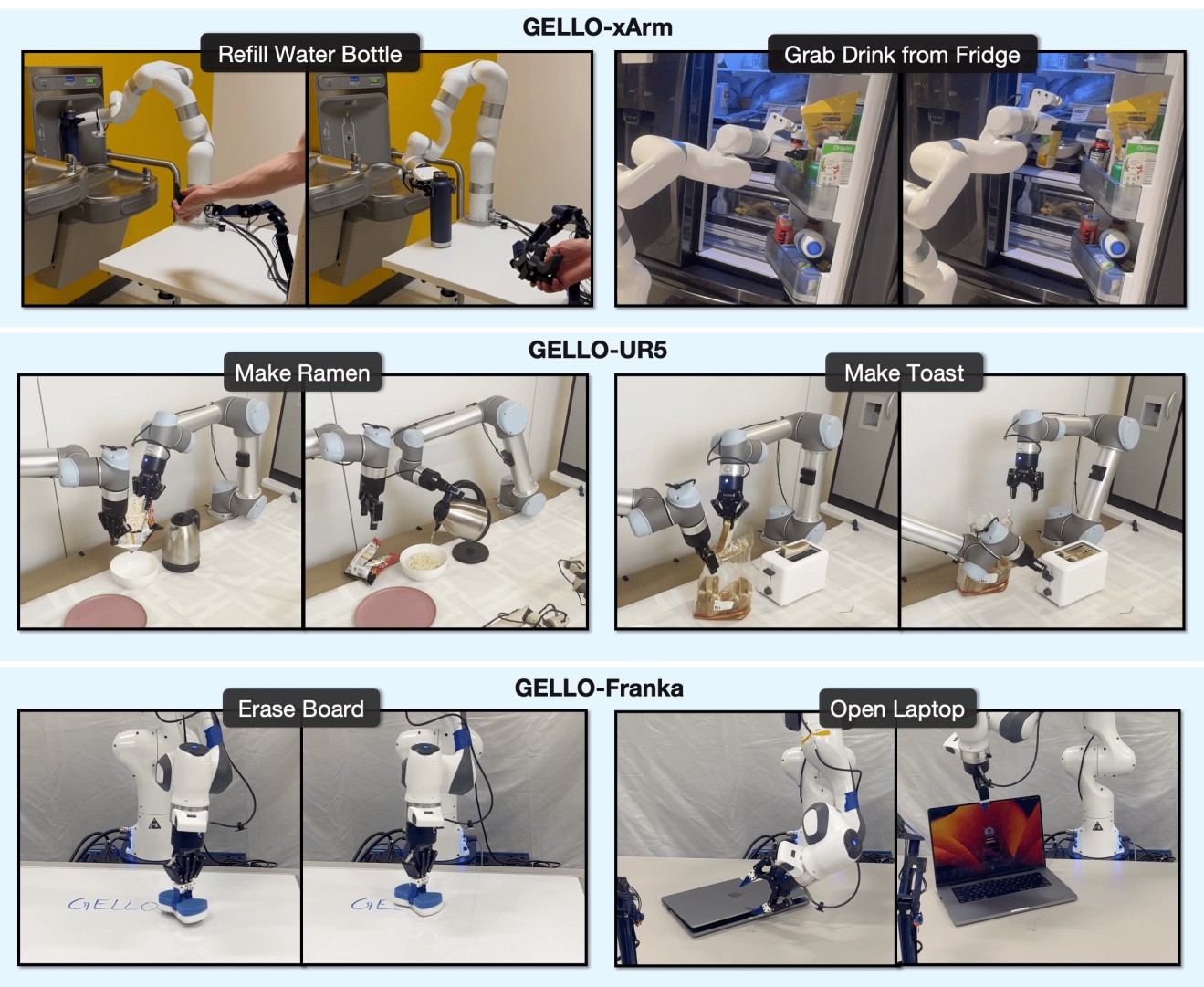

**Fig. 3:** We design and instantiate GELLO for 3 different robot arms. We experiment with GELLO across these three arms on a range of tasks to qualitatively observe the teleportation behavior. GELLO's portable and self-sufficient design enables us to gather data across a wide range of environments.

Following these simple design principles, we instantiate and test GELLO for 3 commonly used robot arms, the Universal Robot UR5, uFactory xArm7, and Franka Panda. Example tasks that we can perform with GELLO on the different robots are illustrated in Figure 3.

## IV. EXPERIMENTS

We conduct experiments to evaluate GELLO as a teleoperation system. Quantitatively, we compare the performance of GELLO with other common low cost teleoperation systems used in the literature across 5 tasks exploring various aspects of manipulation. Qualitatively, we study GELLO through tests on robots from 3 different manufacturers across a variety of manipulation tasks in diverse settings.

### A. User Study Procedure

We conducted a user study involving 12 participants, focusing on bi-manual robot teleoperation using two UR robots to assess the comparative effectiveness of GELLO,

3D mouses, and VR controllers under controlled conditions. An overview of our experimental study is shown in Figure 4, where details of the 5 tasks we study are provided. Control with a 3D mouse or VR controller requires additional tuning of scale parameters that effect how sensitive the controller is to human input. We tune each by testing the device across the 5 tasks, insuring control to be sensitive enough to achieve the fine grain USB insertion tasks while still being responsive enough for quickly traversing across the workspace in the banana hand off task.

Prior to experimenting with the teleoperation tools, each user was granted a brief 6-minute orientation session, introducing them to the basics of the robot and teleoperation itself, as well as the task requirements. To reduce potential biases, no device-specific instructions were given in this orientation, and video demonstrations of the task do not show any particular device. This is then followed by the sequential introduction of the 3 different teleoperation devices. Upon introducing

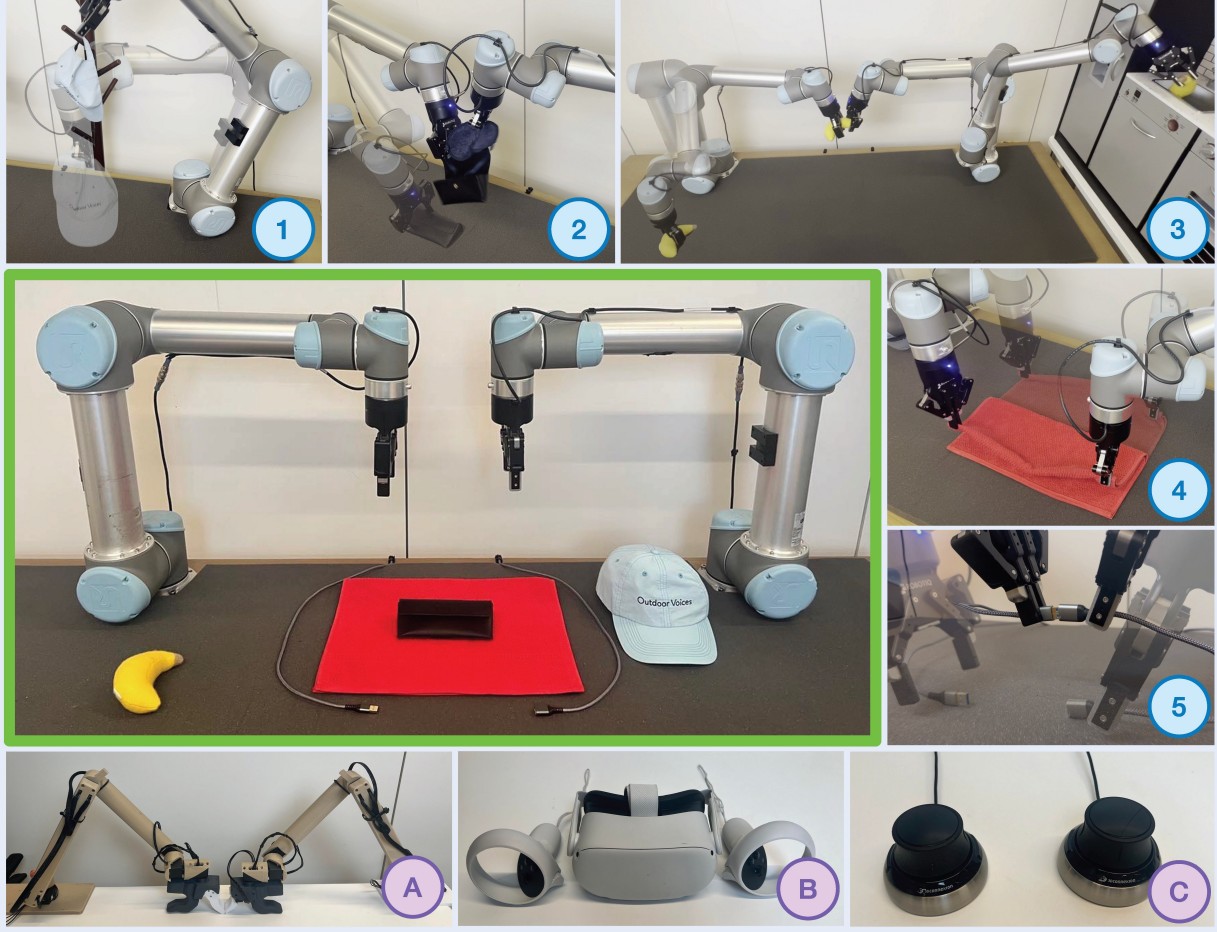

**Fig. 4:** An illustration of the experimental setup for our user study. The primary workspace of our experiments, indicated in green, showcases the scene of a bimanual robot station comprising of two UR5 robots with the complete task suit. The figure details the five tasks which are represented by the numerically labelled frames in blue: (1) Place a hat on a rack, (2) Open a case and fetch the sleeping mask inside, (3) Hand over a banana to the kitchen area, (4) Fold a towel, and (5) Plug in a USB cable. These tasks are designed to explore different teleoperation challenges such as articulated object interaction, large workspaces, deformable objects, and precise insertion. The three different bimanual teleoperation devices are shown by the letter-labelled images: (A) GELLO, (B) VR (Meta Quest2) controllers, and (C) 3D Mice (SpaceMouse). Each user attempts to accomplish the 5 tasks with all 3 teleoperation devices.

a new device, users were given a 5-minute practice phase, allowing them to gain familiarity with the device and its usage. Following the practice phase, the participants begin the task execution phase, with a time limit of 45 seconds for the hat task, and 90 seconds for the remaining tasks. This is repeated for all 3 teleoperation devices. The order in which each participant learns to use the three devices is randomized and given by the experimenters. This eliminates any potential bias from a fixed ordering. Users were instructed to solve the task as best they could while avoiding self-collision or collision with the environment. The robot will be stopped when failure happens and the current task is terminated immediately. We record the task success and failure mode or task completion time as applicable. At the end, we have the user accomplish the same tasks without a robot using their hands.

### B. User Study Results

In Table II, we show the success rate across distinct tasks for each teleoperation device. Using GELLO consistently

**TABLE II:** The task success rate for different teleoperation systems. Ours archives the top success rate across the board.

| Device | Hat | Mask | Banana | Towel | USB | *Avg* |
|--------|-----|------|--------|-------|-----|-------|
| Gello | 0.92 | 0.92 | 1.0 | 0.92 | 0.83 | **0.92** |
| 3D Mice | 0.75 | 0.58 | 0.67 | 0.58 | 0.58 | 0.63 |
| VR | 0.92 | 0.83 | 0.75 | 0.58 | 0.5 | 0.72 |

results in the top success rate. For the simplest task, which only requires controlling a single arm, placing the hat on the rack, GELLO and VR perform comparably. In more complex tasks, such as the banana hand off which requires both arms to maneuver across the large workspace, using the spacemouse or the VR controllers leads to more failures like self-collisions or hitting singularities, suggesting that GELLO offers easier control in more broad use cases. For the task of towel folding, GELLO also shows a large advantage over the other two devices. We hypothesizes that this is because GELLO offers more intuitive control and thus better

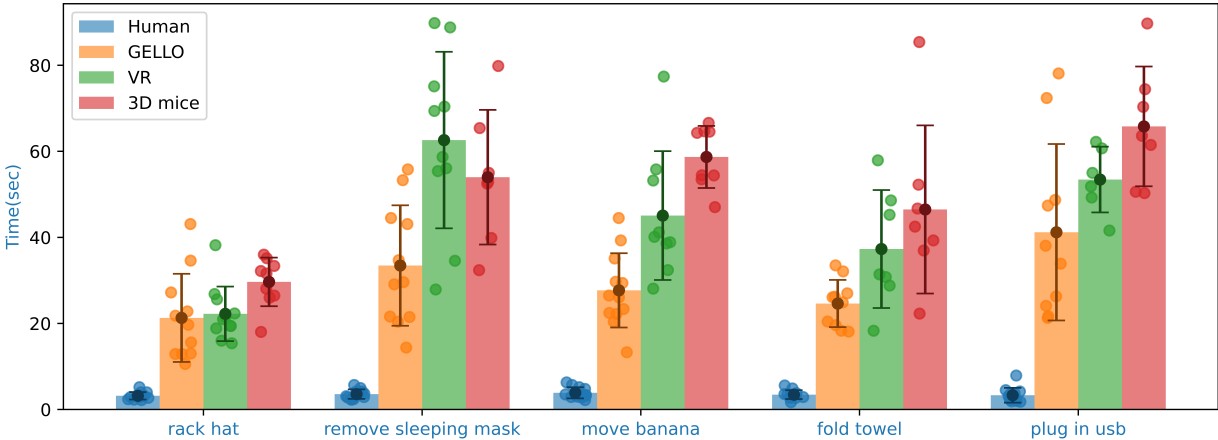

**Fig. 5:** Comparison with other teleoperation systems on the required duration for each task. For every combination of task and system, the average completion time (smaller is better) is plotted only for successful trials. Colored dots indicates the completion time for each user under each task-system pairing. Human, plotted in blue, gives a lower bound on teleoperation completion time, where the user directly accomplished each task with their hands.

coordination between the two arms. Interestingly, for the task of plugging in the USB cable, using GELLO also resulted in a much higher success rate. This is despite VR or spacemouse having the advantage of simplifying the problem by keeping one controller static while focusing their attention on the other, a common strategy users employed. In Table III, we present a breakdown on the different failure modes for each teleoperation system. Our observations suggest that, unlike other devices that operate in Cartesian space, GELLO requires minimal user expertise. This is further corroborated by the lowest timeout count for GELLO in comparison to other teleoperation systems. Furthermore, GELLO's isomorphic joint structure design ensures that teleoperation with GELLO has the least collision risk.

Figure 5 provides a summary of the completion times taken by each device across all five tasks, given successful task execution. Utilizing GELLO results in consistently faster completion times. This not only signifies that GELLO is easier to use with a higher success rate but also, indicates its efficiency; faster completion times would enable users to achieve more successful operations in a given time frame.

### C. Teleoperation System Capabilities

We further demonstrate the capabilities of GELLO on more challenging manipulation tasks, including in real-world environments. We show some of them in Figure 3 and put more videos on our project website. These tasks include contact-rich tasks, long-horizon tasks and challenging bimanual coordination tasks. Tasks like filling water bottles require a certain payload on the robot arm and would be difficult to perform for smaller arms to perform, such as the ViperX arm used in the Aloha system [11]. GELLO is also effective for 7 DOF arms, such as the Panda and xArms, despite the extra degree of freedom. We find that the kinematically equivalent structure enables a user to directly manage the arm's null space if required, which can be advantageous when operating in cluttered spaces.

**TABLE III:** The failure count for each failure mode aggregated across all 5 tasks. Each ✗ indicates a single failure of that type from our trials. The "Other" category captures all other irrecoverable task failure modes such as dropping the working item outside the reach of the robots.

| Failure Mode | Gello | 3D Mice | VR |
|---|---|---|---|
| Timeout | ✗ | ✗✗✗✗✗✗✗✗ | ✗✗✗✗ |
| Self Collision | ✗ | ✗✗✗✗✗ | ✗✗✗✗✗ |
| Env Collision | ✗✗✗ | ✗✗✗✗✗✗ | ✗✗✗✗✗✗ |
| Other | | ✗✗ | ✗ |

## V. DISCUSSION

Due to limited output torque of the motors used, GELLO does not provide force feedback to the users, which limits GELLO's capabilities when teleoperating for more contact-rich tasks. We made this compromise to keep GELLO low-cost, more accessible, and more applicable to all robot arms, as bilateral devices also require force sensing capability for the target robot. However, we hope to incorporate this as an optional capability in the future for more advanced usage.

Our user study is limited to inexperienced users who are only briefly taught about teleoperation and who only practiced for a limited time. Additional training can significantly improve the user's proficiency in using teleoperation devices and we leave such study to future work.

In this paper, we introduced GELLO, a general low-cost teleoperation platform for manipulation. Our results demonstrate its effectiveness through a user study on teleoperation with a bi-manual robot system using two UR5s. To demonstrate versatility and make GELLO more accessible, we design GELLO for 3 robots. We hope GELLO will lower the barrier to collecting large and high-quality demonstration datasets, and thus accelerate progress in robot learning.

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
