# OpenReview forum: "GELLO: A General, Low-Cost, and Intuitive Teleoperation Framework for Robot Manipulators"
_robot-learning.org/CoRL/2023/Workshop/TGR — CoRL 2023 Workshop TGR Oral_

### Official Review · Reviewer_jTka · 2023-10-19

**Rating:** 8
**Confidence:** 3

**Review:**

This paper introduce GELLO, a low-cost, intuitive teleoperation framework for robot manipulators, designed to be user-friendly and affordable. The GELLO system has the same kinematic structure as the target arm, leveraging 3D-printed parts and off-the-shelf motors. It allows efficient demonstration collection for dexterous and complex manipulation. This work opens up a more promising and scalable solution toward collecting human demonstration of diverse dexterous manipulation, which is highly relevant with the workshop.

---

### Decision · Program_Chairs · 2023-10-20

**Decision:**

Accept (Oral)

**Comment:**

Great paper and very cool device for scaling up real world data collection!